# A Study of Structural Change during In Vitro Digestion of Heated Soy Protein Isolates

**DOI:** 10.3390/foods8120594

**Published:** 2019-11-20

**Authors:** Tian Tian, Fei Teng, Shuang Zhang, Baokun Qi, Changling Wu, Yan Zhou, Liang Li, Zhongjiang Wang, Yang Li

**Affiliations:** 1College of Food Science, Northeast Agricultural University, Harbin 150030, China; tiantian@neau.edu.cn (T.T.); tengfei@neau.edu.cn (F.T.); szhang@neau.edu.cn (S.Z.); qibaokun@neau.edu.cn (B.Q.); wuchangling@neau.edu.cn (C.W.); zhouyan4825@neau.edu.cn (Y.Z.); liliang@neau.edu.cn (L.L.); wzjname@126.com (Z.W.); 2Department of Food Science, Cornell University, Ithaca, NY 14853-7201, USA

**Keywords:** soy protein isolate, in vitro digestion, heat treatment, structure

## Abstract

Use of soy protein isolate (SPI) as the encapsulating material in emulsions is uncommon due to its low solubility and emulsification potential. The aim of this study was to improve these properties of SPI via heat treatment-induced modifications. We modified SPI under various heating conditions and demonstrated the relationship between structure and in vitro digestibility in simulated gastric fluid by means of Sodium Dodecyl Sulphide-Polyacrylamide Gel Electrophoresis (SDS-PAGE) and Raman spectroscopy. It was found that the degree of hydrolysis (DH) of SPI increased and then decreased upon increasing exposure to heat. Different subunits of conglycinin were digested and degraded by pepsin. Heat treatment improved digestion characteristics that would reduce e the unnecessary loss of protein, offering potential for the efficient delivery of nutrients in nanoemulsions. These results could have significant relevance for research groups that are interested in the biological interactions and activity of functional SPI.

## 1. Introduction

Soybeans are one of the main sources of plant protein for human and animals, largely because they have higher protein content (approximately 40%) than other grains and beans [1]. Therefore, soybeans have become one of the most important economic and commercial agricultural products. Soy protein isolate (SPI) is an important ingredient with many applications in the food industry because of its physicochemical characteristics, nutritional value, and low cost of production. Due to its potential emulsification, gelation and hydration, SPI is widely applied in the food industry in beverages, nutritious food, fermented products and more. Additionally, SPI can reduce serum cholesterol and prevent heart and cerebrovascular diseases in the form of foods on the market (e.g., functional foods, pre and probiotics, food supplements, botanicals) and in pharmaceuticals [2,3,4]. The bioavailability of nutrients affects its efficacy as a disease prevention agent. SPI has been suggested as an oil–water emulsion nanoparticle delivery system [5,6]. When used as an emulsifier, amphophilic SPI can adsorb to the surface of oil droplets, stabilizing them. However, the application of SPI is still limited by its poor solubility, low emulsifying capacity and poor bioavailability [7].

Heat treatment is a traditional processing method, which affects the functional properties and vital activities of proteins. Generally, heat treatment can expand the rigid structure of a protein to form a flexible structure that is easy to digest. In addition, the heat treatment reduces the content of the *β*-sheet structure and imparts better biological activity and functional properties to the protein [8]. Several researchers have investigated the structure and functional properties changes of SPI, glycinin (11S), and *β*-conglycinin (7S) during heat treatment [9,10,11,12,13]. Hu et al. [14] found that the structure of glycinin was maintained when heated at the temperature range of 20–60 °C. However, heating above 80 °C significantly decreased the content of *β*-sheets and increased the content of random coils [15,16]. Natarajan et al. [17] revealed that the first denaturation temperature of *β*-soy conglycinin was 65 °C. When the temperature of the protein rose to 75 °C, the content of *β*-sheet structure no longer increased. Prior study of the conformational change of glycinin thermogels revealed that glycinin is mainly composed of *β*-sheet and irregular structures in its natural state, while the content of *β*-sheet in the state of gel was markedly decreased at 75 °C [18]. Thus, heat treatment plays an important role in the structure and nutritional value of soybean products.

Despite these successes, little attention has been devoted to protein digestion. The purpose of this study was to investigate the effects of heat treatment on the structural and digestive properties of SPI. We studied how these properties changed as a function of heating temperature (70–100 °C) and heating time (10–60 min), in order to obtain a better understanding of physicochemical effects of heat treatment on SPI which may lead to improved applications in the food industry.

## 2. Materials and Methods

### 2.1. Materials and Chemicals

Soybeans for this study were purchased from Hei Longjiang Agriculture Co., Ltd. (Harbin, Hei Longjiang, China). Pepsin with an activity was 3000 U/mg was purchased from Beijing Wohai Technology Ltd. (Beijing, China). An SDS-PAGE gel preparation kit was purchased from Beijing Suo Laibao Technology Co., Ltd. (Beijing, China). All other chemicals were of analytical grade and were procured from Tianjin Chemical Reagent Co. (Tianjin, China).

### 2.2. Preparation of Soy Protein Isolates

SPI was prepared according to the following process, as previously described [19]. Soybeans were peeled, crushed, and sifted through a 100-mesh screen. Flours were degreased three times with n-hexane 1:6 (*w*/*v*) in a 37 °C water bath. Defatted soy flours were mixed with deionized water at a ratio of 1:10 (*w*/*v*) with the pH adjusted to 8.0 (2 mol/L NaOH). The mixture was stirred continuously for 2 h in a 50 °C water bath. Afterward, the solution was centrifuged at 10,000 rpm for 30 min at 4 °C. After centrifugation, the pH of supernatant was adjusted to 4.5 using 2 mol/L HCl, and centrifuged at 6500 rpm for 30 min at 4 °C. The precipitate was washed with deionized water for 48 h at 4 °C and neutralized to pH 7.0 using 2 mol/L NaOH, followed by dialysis, and freeze-drying. The final product was SPI. All steps were performed at room temperature. The preparation of SPI is shown in Figure 1.

### 2.3. Heat Treatment of SPI

The protocol for heat treatment of SPI was based on our previous research with slight modifications [20]. Five g of SPI were first dissolved in 100 mL of phosphate buffered saline (PBS, 0.1 M, pH 7.4). The aqueous dispersion was stirred in a sealed glass tube, and heated from 70 to 100 °C for 15 min, or heated at 85 °C for 10 to 60 min. The heated sample (HSPIs) was centrifuged at 9500 rpm force for 20 min at 4 °C. After removing the insoluble compounds, the sample was cooled immediately in an ice bath for further experiments.

### 2.4. In Vitro Pepsin Digestion of HSPIs

Pepsin digest of HSPI was conduct according to Chen et al. [21] with slight modifications. Simulated gastric fluid (SGF) consisted of 3.2 mmol purified pepsin (3000 units/mg protein, pH 1.2) containing 35 mmol NaCl. Pepsin solution was added drop-wise into SGF while vortexing for 5 min. The resulting solution was placed on ice. The concentrations of all test samples were 5% w/w of the HSPIs for SGF digestion assay. The in vitro gastric model consisted of a conical flask (100 mL) containing 10 mL of SGF-pepsin maintained at 37 °C with continuous shaking at 95 rpm. Aliquots (10 mL) were withdrawn into beakers 1 h at intervals during incubation, and 75 μL of 200 mmol Na_2_CO_3_ (pH 11.0) was added to each mixture to stop the reaction by neutralization. The digestion was replicated in triplicate. After freeze-drying, the samples were stored at 4 °C until use.

### 2.5. The Degree of Hydrolysis Measurement

The degree of hydrolysis (DH) of soy protein was measured based on the ortho-phthalaldehyde (OPA) method [22,23]. The DH was determined from the percentage of peptide bonds that are hydrolyzed by the protein. Each peptide bond is hydrolyzed with a free amino group released. The free amino group, and OPA generate a yellow complex. Diluted HSPIs digests (400 μL) were mixed with 3 mL OPA reagents for 2 min, and the absorbance was measured at 340 nm by SP-721 UV spectrophotometer (Beijing Purkinje General Instrument Co. Ltd., Beijing, China). Free amino groups in digests of HSPIs were expressed as serine amino equivalents (serine NH_2_equiv). DH (%) was calculated by the following equations:(1)DH(%)=hhtot×100,
(2)$$h=\frac{SerineNH_{2}-\beta }{\alpha },$$
where *β* equals to 0.342 mequv/g, *α* equals to 0.970, and *h_tot_*, equals to 7.8 mequv/g for soy protein

### 2.6. Raman Spectroscopy

Raman spectroscopy is a direct and non-invasive technique which has been used to study the structure of SPI [24,25]. Each spectrum of the sample was collected at 785 nm laser excitation wavelength, 300 mW laser power, four scans, and 60 s exposure time. Baseline calibration was made for the average spectral data of samples scanned from Raman spectrophotometer (Renishaw, Gloucestershire, UK), and the phenylalanine band at 1003 cm^−1^ was standardized. The Raman spectra of each sample were analyzed in the region of 400–3100 cm^−1^. Each sample was scanned three times. After calculating the average value, the Raman spectra were drawn. The relative standard deviation was less than 5%.

### 2.7. Sodium Dodecyl Sulphide-Polyacrylamide Gel Electrophoresis

Sodium dodecyl sulphide-polyacrylamide gel electrophoresis (SDS-PAGE) was performed according to the method of Laemmli et al. [26]. Preparation for the 5% stacking gel and 12% separating gel was according to the specification provided by Suo Laibao Technology (Beijing, China). Tweenty μL protein samples or their digestive products were mixed in a buffer (1:1, *v*/*v*) containing *β*-mercaptoethanol in an equal volume and then denatured inboiling water for 5 min. The samples were cooled at room temperature, and then 10 μL aliquots of the samples (5 mg/mL) were loaded into the gel. Electrophoresis was carried out at 80 mV in the stacking gel and then separated in the gel under 120 mV. After about 3 h of electrophoresis, coomassie brilliant blue was stained with R-250 for 12 h until the dye reached the bottom of the gel. After electrophoresis, the gel was stained with 0.05% coomassie brilliant blue R-250 solution until a clear background appeared. We used a Molecular Imager Gel Doc (Bio-Rad Laboratories, California, CA, USA) to capture the gel.

### 2.8. Determination of Molecular Weight

Molecular weight distribution was determined according to the method of Yang et al. [27]. The samples were determined on an Agilent High Performance Liquid System and HiLoad 16/60 Superdex 200 prep grade pre-packed preparative columns with an Agilent UV detector set to 280 nm. Sample was eluted with mobile phase (0.02 M phosphate buffer containing 0.25 M NaCl (pH 7.2)) at a flow rate of 1 mL/min. A calibration curve of molecular weight was plotted from the average elution volume of the following standards: Cytochrome C (12,384 Da), aprotinin (6500 Da), vitamin B12 (1855 Da), oxidized glutathione (612 Da), and Glycine (75 Da) (Sigma Chemical Co., St.Louis, MO, USA) were taken to make reference curve (*r* = 0.9955). A regression equation was established for the relative molecular mass (Mw) and elution volume (x) of Superdex 200 and Sephadex G-75. Aliquots of 20 μL samples were injected into the column. The molecular mass was estimated based on the elution time against those of molecular weight markers. The relative content of each peptide fraction was expressed as the percentage area of its chromatogram peak.

### 2.9. Statistical Analysis

All reported values were performed in triplicate, and the results were reported as means and standard deviations. The results were analyzed by analysis of variance (ANOVA). Data was analyzed using Origin 8.5 software, OMNIC software package, and PeakFit 4.12 software. All statistical calculations were performed using a commercially available computer software package (SPSS software, Version 22, IBM, Armonk, NY, USA).

## 3. Results

### 3.1. Effect of Heat Treatment on DH of SPIs In Vitro

The DH is defined as the proportion of cleaved peptide bonds in a protein hydrolysate [28]. The effect of preheat treatment on DH of SPIs in vitro is shown in Figure 2. The DH of HSPIs increased with heating temperature until 85 °C and then decreased. Similarly, The DH of HSPIs initially increased with increasing heating time but then decreased (>30 min). These observations suggested that heat treatment partly denatured SPI and exposed the side chain reactive groups of SPI, which facilitated protease binding [29]. The resulting enzymolysis causes the hydrogen bond to break, exposing some embedded hydrophobic residues. This is beneficial to the diffusion of the oil–water interface, reducing interfacial tension and improving emulsifying ability. Our previous research showed that SPIs with a high DH had higher solubility attributed to the reduction in molecular weight (MW) and increase in ionizable amino and carboxyl groups of SPI. This observation is consistent with the findings of Liu et al. [30], who also found that heat treatment disrupted the protein tertiary structure and increased the DH. However, when treated at a higher temperature, the DH decreased. This was because HSPIs aggregated and precipitated, prohibit the binding of enzyme to SPI. According to our previous research, 7S globulin denatured at 80 °C, and 11S globulin denatured at 90 °C [31]. It can be inferred that whereas denaturation of 7S globulin increased the DH of HSPIs in vitro, denaturation of 11S globulin inhibited the digestion of HSPIs. The 11S globulin denatured at 90 °C and interacted with denatured 7S globulin forming thermal aggregations, with part of the enzyme site embedded on the surface of the protein. This is detrimental to the combination of proteases and DH reduction. These results are consistent with the findings of Kondjoyan et al. [19]. Thus, heat treatment can significantly increase the DH of SPI in vitro, and SPI’s greatest susceptibility to enzymatic hydrolysis was obtained after heating at 85 °C for 30 min.

### 3.2. Protein Structure Was Determined by Raman Spectra

Analysis of the Raman spectra of proteins allow the unambiguous identification by spectral data base search algorithms [32]. The Raman spectra of HSPIs are shown in Figure 3.

#### 3.2.1. Secondary Structure

Amide I and III are the most important bands in determining the secondary structure of proteins in several distinct vibration modes of the –CO–NH– amide [33]. The main Raman characteristic peaks of the amide I band of the native SPI and HSPI were between 1654 and 1662 cm^−1^, indicating that the advantages of α-helices and random coil structures were presented (Figure 3). The amide III band at around 1246 cm^−1^ (random coil) and 1275 cm^−1^ (α-helix) was attributed to the in plane bending of N–H and stretching of C–N. The absorption at 1305 cm^−1^ in the amide III region could be derived from a *β*-turn.

Figure 3 shows a slight shift to lower frequencies of the intensity maximum of this band (1660 cm^−1^) due to the heating process, indicating an increment in the α-helical structure due to thermal treatment. It has been reported that the secondary structure of digested SPIs were unfolded with heat treatment, and that the protein changes from ordered to disordered structures. Based on our previous reports, heat treatment resulted in an increase in the content of α-helix and *β*-turn structure, and a decrease in the content of *β*-sheet structures [34]. During the in vitro digestion process, *β*-conformations are prone to be hydrolyzed by pepsin and converted into an unordered structure, followed by the digestion of the α-helix and unordered structure [16]. The content of *β*-sheets and unordered structures of HSPIs decreased at a low temperature, while the content of *α*-helixes and *β*-turns increased generally. It has been reported that heat treatment induced a self-reassembly from *β*-sheet to *α*-helix and *β*-turn structures [35]. Lee et al. [9] found that the *β*-sheet structure was mainly stabilized by hydrogen bonding between the carbonyl group (–CO) and the amino group (–NH), and that the *β*-sheet structure tended to be embedded in the polypeptide chain, which inhibited the digestion of SPIs. Thus, there was a significant negative linear correlation between the content of *β*-sheets and the degree of hydrolysis. Heat treatment generally increased the content of *α*-helixes and decreased the content of *β*-sheets of digested SPIs, suggesting increased digestion of *β*-sheets. The *α*-helix content of digested SPIs first increased and then decreased with increased temperature, while the *β*-sheet content generally increased. It was reported that the content of *α*-helix decreased because of denaturation of 7S globulin, while the increased *β*-sheet structure of digested HSPIs was associated with aggregates formed by the *α*’- and *α*-7S subunits [35,36]. The increased *β*-turn contents of digested HSPIs may be related with the intermediate aggregate of AB subunit enhanced by increasing heat treatment [9,37].

However, the SPI structure denatured and formed aggregations with the extension of heating time, and the frequency of ordered secondary structurse increased. The α-helix and random structure content of digested SPIs increased and then decreased with increasing heating time, while the *β*-sheet content first decreased and then increased.

#### 3.2.2. Changes of Protein Tertiary Structure

Raman spectra of native SPIs and HSPIs could provide information on the tertiary structure of proteins from their tryptophan, tyrosyl doublet, and aliphatic hydrophobic residues [33,37,38].

The tyrosine bimodal ratio (I850/I830) is a Fermi resonance that can be used to monitor the microenvironment around tyrosine residues [39]. Figure 3 shows the Raman bands near 850 cm^−1^ and 830 cm^−1^ representing the respiratory vibration of the benzene ring of the tyrosine residue, and the octant residue extra-fold bending frequency doubling, respectively. When the I850/I830 ratio ranged from 1.25–1.40, the tyrosine residues were exposed to the aqueous or polar environment. On the contrary, when the ratio ranges from 0.7 to 1.0, even as low as 0.3, the tyrosine residues were buried in a hydrophobic environment. As can be seen from Table 1, no significant change in I850/I830 ratio was observed between digestion products of native SPIs and HSPIs. Similar findings were reported by Zhang et al. [40], specifically that the secondary structure contents (*α*-helix, *β*-sheet, and random coils) of SPI did not change significantly (*p* > 0.05) after heat treatment.

The band at 760 cm^−1^ is known to represent monitor tryptophan residues, which indicates that the stretching vibration of the tryptophan residues has changed [41]. The decrease of the band strength near 760 cm^−1^ indicated that more hydrophobic resides were exposed to the polar aqueous solution [33,37]. Table 1 shows the effect of heat treatment on the SPI tryptophan side chain. Li et al. [37] found that the tryptophan residue transformed from "buried" to "exposed". With increasing temperature, the intensity of tryptophan in the digestion products of HSPIs first increased and then decreased. This suggested that most exposed tryptophan residues might be hydrolyzed. The tyrosyl double bond ratio (I850/I830) can be used to characterize the microenvironment around the tyrosine residue. We found that the structure showed that the difference after heat treatment was not significant [42].

### 3.3. Effect of Heat Treatment on Subunit Composition of SPI In Vitro by SDS-PAGE Profile

SDS-PAGE patterns of native SPIs and HSPIs digested with pepsin for 1 h are shown in Figure 4. Untreated and heated SPIs profiles presented a typical SDS-PAGE representation of individual subunits of *β*-conglycinin (7S) (*α*, *α*’, and *β* subunits) and glycinin (11S) (acidic (A) and basic (B) subunits).

The cluster of protein bands with molecular weights of the 11S fractions were approximately 80, 75, and 50 kDa. The 36 kDa subunit was acidic A3 polypeptide and the group of polypeptides around 34 kDa were a major group of acidic polypeptides (A1, A2, and A4). The molecular weights of the 11S fraction were approximately 15 kDa [43].

Figure 4 shows that the bands corresponding to 11S globulin were digested; however the bands corresponding to 7S globulin remained. Several new bands of 11–15 kDa were observed corresponding to hydrolyzed peptides. This suggests that pepsin decreased acidic and basic subunits of 11S globulin in native SPI. Similarly, Amigo et al. [32] pointed out that the electrophoretic bands of the *α* and *β* subunits of 7S were observed after digestion by pepsin, but the basic subunits of 11S globulin of natural SPI were not detected.

During pepsin digestion, the intensities of the protein bands corresponding to 7S globulin fractions decreased and many peptide bands appeared around 11–15 kDa due to protein hydrolysis. When the temperature was higher than 80 °C, the 7S globulin was denatured and easily digested by pepsin [20]. Although no significant differences (*p* > 0.05) were observed between any of the protein bands corresponding to 11S globulin fractions, this might be due to the difficulty in digesting 11S globulin of HSPIs, or the competition of 7S globulin and 11S globulin on pepsin [33,36]. A new band appeared at 100 °C, likely because heat treatment caused 7S globulin and 11S globulin to produce aggregates that were not easily digested. With prolonging heat treatment, the intensities of the protein bands corresponding to *α*’-, *α*-, and *β*-7S globulin fractions were gradually decreased, and the *β*-subunit was not significantly digested compared with the α-subunit. Thus, protein digestion is a continuous process. Heat treatment produced new protein bands, which might be from the digestion products of 7S globulin and 11S globulin or might otherwise be insoluble aggregates produced by HSPIs [44].

### 3.4. Molecular Weight Distribution after In Vitro Digestion

#### 3.4.1. Effects of Heating Temperature on Digestion Characteristics of SPI

The molecular weight and distribution of the SPI digestion solution at different temperatures during the simulated digestion process was determined using a gel chromatography column (Sephadex G-75) [27]. The molecular weight distribution of the digestion products is shown in Table 2 and Table 3.

With increasing temperature, the content of high molecular weight (HMW) proteins (>100,000 Da) in the digestion products significantly increased (*p* < 0.01), while the content of medium molecular weight (MMW) peptides (10,000–100,000 Da) decreased and small molecular weight (SMW) (3000–10,000 Da) increased. No significant difference was observed in the content of low molecular weight (LMW) peptides (<3000 Da). HMW protein was related to the formation of heat-induced aggregation, which hindered the effect of digestion on the protein. The increase in HMW protein content was not significant as heating increased, as well as the MMW digested to SMW [45].

#### 3.4.2. Effect of Heating Time on the Digestion Characteristics of SPI

As heating time was prolonged, the content of HMW and SMW increased, while MMW decreased (Table 2). However, no significant differences (*p* > 0.05) were observed in the content of SMW peptides (<3000 Da). The macromolecular proteins produced by digesting heat-treated protein increased significantly with increasing heat. Protease digested the medium molecular weight peptides (10,000–100,000 Da) to small molecular weight peptides (3000~10,000 Da), as previously described [22,46,47].

Together, we found that HMW peptides (>100,000 Da) increased with heating temperature and time. However, MMW peptides (10,000–100,000 Da) and peptides below 3000 Da decreased with the increase of heating temperature and time. Additionally, the medium and small molecular weights ranging from 3000 to 10,000 Da were slightly increased relative to the untreated protein. This is because heat treatment induced protein denaturation, and the tight structures loosened to form soluble and insoluble aggregates, decreasing solubility and the ability of enzymes to digest the protein. Also, Xu et al. [48] discovered that heat treatment promoted the disruption of protein crystals and dissociation of double helical structures in the amorphous region, which can facilitate enzymatic hydrolysis within protein granules.

As shown in Table 3, although there was no correlation between SMW, LMW, and DH, the HMW content was significantly positively correlated with DH (*p* < 0.01, *r* = 0.88), and MMW content was significantly negatively correlated with DH (*p* < 0.01, *r* = −0.83). The results showed that with increasing temperature, the digestibility of soybean protein decreased, the content of high molecular weight peptide aggregates increased, and the content of medium and small molecular weight peptide increased. There was no correlation between MMW, SMW, LMW, and DH at different heating times; however HMW content showed a significant positive correlation with DH (*p* < 0.05, *r* = 0.82), indicating that protein digestibility decreased and the peptide content of the macromolecules in the digested products increased with extended durations of heating. This indicates that the formation of HMW protein aggregates is an important factor affecting the digestion of SPI.

## 4. Conclusions

Food containing SPI is often affected by heat treatment throughout the course of production and processing, and heat treatment has a significant impact on the solubility, emulsification potential, and stability of SPI. In this paper, heat-treated SPI was studied by in vitro digestion assays in order to solve the structural changes of SPI in the human digestive system. Here, we have demonstrated that digestion of SPIs is greatly affected after exposure to heat. After the heating process, the subunit component of *β*-conglycinin of SPI was easier to digest and degrade compared with native SPIs. The digested products had a higher α-helix structure and a lower *β*-sheet structure. In summary, functional protein peptides with digestible, functional and antioxidant properties can be obtained by heat treatment. Moreover, heating enhanced both the emulsification and digestive use rate of SPI, indicating that the bioavailability of SPI can be improved. It can also effectively replace the nutritional and health effects of animal protein on human body. This study may provide valuable information on potential new avenues of application for SPI as a novel protein resource in the food industry.

## Figures and Tables

**Figure 1 foods-08-00594-f001:**
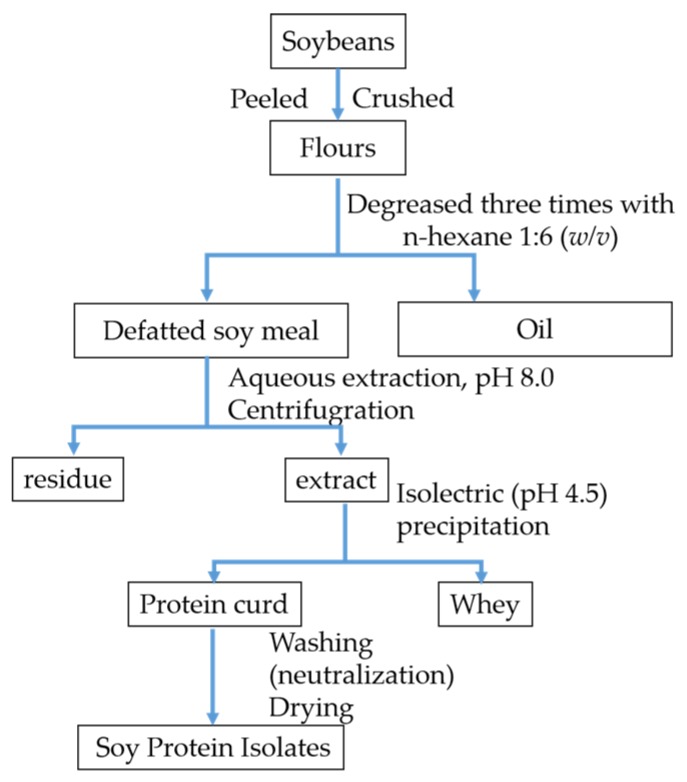
Flow chart of soybean protein isolate preparation.

**Figure 2 foods-08-00594-f002:**
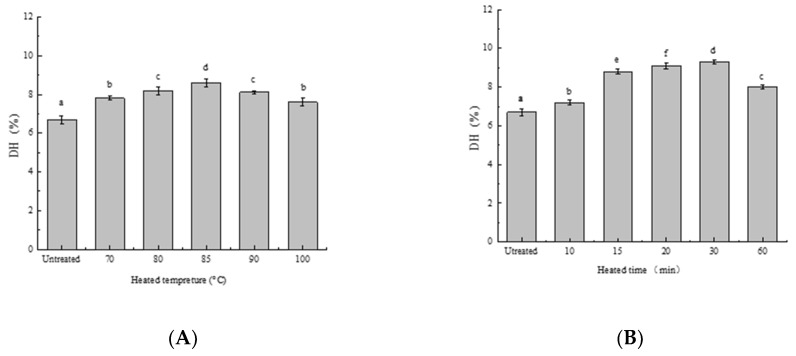
The Degree of Hydrolysis Measurement of SPI in vitro after treatment at different (**A**) temperatures and (**B**) times. The data with different letters are significantly different (*p* < 0.05).

**Figure 3 foods-08-00594-f003:**
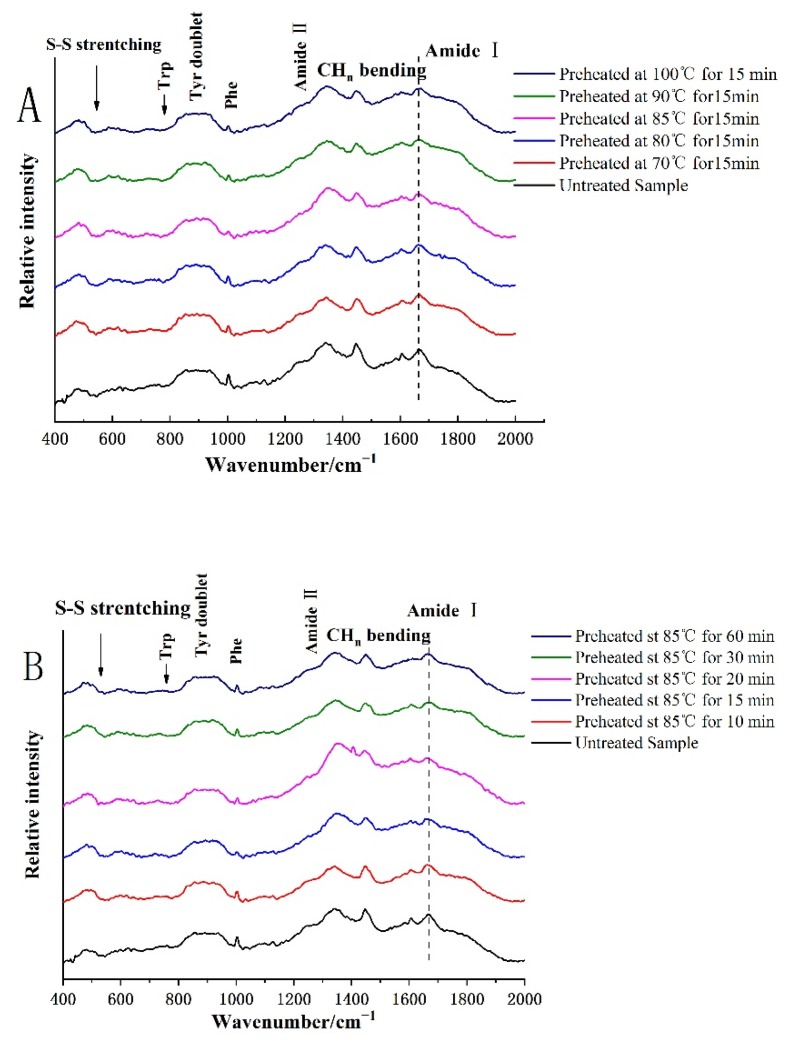
Raman spectra of in vitro digestion products of SPI treated at different (**A**) temperatures and (**B**) times.

**Figure 4 foods-08-00594-f004:**
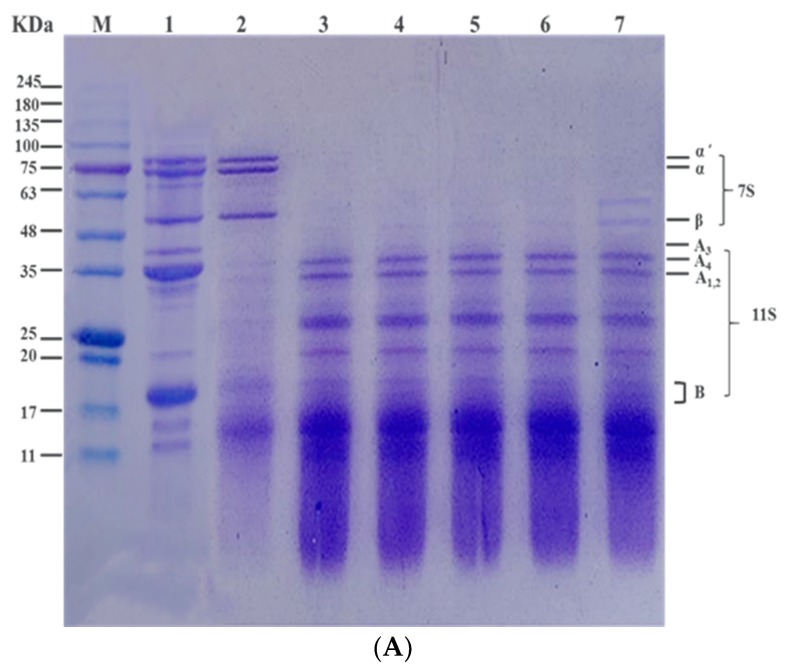
SDS-PAGE electrophoretic profiles of various formulations during the in vitro digestion process. (**A**) M. Standard molecular mass; 1. Electrophoresis is bands for samples without pepsin; 2-7. SPI at different temperatures: untreated SPI and pretreated at 70, 80, 85, 90 and 100 °C for 15 min. (**B**) M. Standard molecular mass; 1. Electrophoresis bands for samples without pepsin; 3–7. SPI at different times: untreated SPI and pretreated for 10, 15, 20, 30 and 60 min at 85 °C.

**Table 1 foods-08-00594-t001:** Contents of the secondary structure prediction by Amide I bands and Fermi resonance ratio I_850_/I_830_ and the number of tyrosine residues that are exposed/buried in heated SPI.

Heat Treatment	Sencodary Structure (%)					
	α-Helix (%)	*β*-Sheet (%)	*β*-Turn (%)	Random Coil (%)	I_850_/I_830_	I_760_/I_1003_
Untreated	13.69 ± 0.03 ^a^	55.68 ± 0.01 ^e^	23.70 ± 0.10 ^e^	6.93 ± 0.03 ^b^	1.02 ± 0.01 ^a^	0.95 ± 0.04 ^a^
70 °C,15 min	17.77 ± 0.02 ^b^	51.41 ± 0.40 ^a^	25.98 ± 0.01 ^e^	4.84 ± 0.02 ^a^	1.01 ± 0.00 ^a^	0.98 ± 0.03 ^c^
80 °C,15 min	20.38 ± 0.01 ^d^	52.48 ± 0.20 ^b^	22.27 ± 0.02 ^c^	4.87 ± 0.01 ^a^	1.01 ± 0.02 ^a^	0.99 ± 0.02 ^d^
85 °C,15 min	18.37 ± 0.20 ^c^	54.78 ± 0.01 ^d^	19.30 ± 0.20 ^a^	7.55 ± 0.03 ^d^	1.02 ± 0.02 ^a^	0.98 ± 0.03 ^c^
90 °C,15 min	18.34 ± 0.02 ^c^	54.42 ± 0.10 ^c^	19.47 ± 0.02 ^a^	7.77 ± 0.01 ^e^	1.02 ± 0.02 ^a^	0.98 ± 0.01 ^c^
100 °C,15 min	17.77 ± 0.20 ^b^	54.53 ± 0.01 ^cd^	20.45 ± 0.03 ^b^	7.25 ± 0.10 ^c^	1.01 ± 0.00 ^a^	0.97 ± 0.03 ^b^
Untreated	13.69 ± 0.03 ^A^	55.68 ± 0.01 ^E^	23.70 ± 0.10 ^E^	6.93 ± 0.03 ^B^	1.02 ± 0.01 ^A^	0.95 ± 0.04 ^A^
85 °C,10 min	14.19 ± 0.10 ^B^	55.80 ± 0.10 ^F^	25.23 ± 0.20 ^F^	4.78 ± 0.03 ^A^	1.01 ± 0.02 ^A^	0.97 ± 0.04 ^B^
85 °C,15 min	18.37 ± 0.20 ^E^	54.78 ± 0.01 ^D^	19.30 ± 0.20 ^A^	7.55 ± 0.03 ^C^	1.01 ± 0.02 ^A^	0.99 ± 0.02 ^C^
85 °C,20 min	18.72 ± 0.20 ^F^	52.58 ± 0.01 ^B^	20.66 ± 0.02 ^C^	8.04 ± 0.01 ^D^	1.01 ± 0.03 ^A^	0.98 ± 0.03 ^BC^
85 °C,30 min	18.31 ± 0.01 ^D^	50.13±0.01 ^A^	22.00±0.10 ^D^	9.56 ± 0.03 ^F^	1.01 ± 0.03 ^A^	0.97 ± 0.04 ^B^
85 °C,60 min	17.35 ± 0.02 ^C^	54.44 ± 0.30 ^C^	20.06 ± 0.02 ^B^	8.15 ± 0.03 ^E^	1.01 ± 0.03 ^A^	0.97 ± 0.02 ^B^

Note: Normalized intensities of the I760/I1003 (tryptophan) band, tyrosyl doublet at I850/I830. Data represent mean ± standard deviations. Different letters in the same column respresent a significant difference between samples (*p* < 0.05). Arrange all the averages in descending order, and use the letter “a” and “A” on the minimum average.

**Table 2 foods-08-00594-t002:** Protein molecular distribution as a function of digestion SPIs at different heat treatment.

Molecular Mass Distribution/Da	>100,000(%)	10,000~100,000(%)	3000~10,000(%)	<3000(%)
Untreated	54.26 ± 0.10 ^a^	38.94 ± 0.10 ^e^	1.31 ± 0.10 ^a^	5.49 ± 0.20 ^c^
70 °C,15 min	65.31 ± 0.20 ^b^	20.55 ± 0.20 ^d^	7.87 ± 0.20 ^b^	6.27 ± 0.10 ^d^
80 °C,15 min	69.57 ± 0.30 ^d^	18.14 ± 0.10 ^c^	8.19 ± 0.10 ^bc^	4.10 ± 0.10 ^a^
85 °C,15 min	69.02 ± 0.10 ^e^	18.46 ± 0.20 ^c^	8.44 ± 0.20 ^c^	4.08 ± 0.20 ^a^
90 °C,15 min	68.51 ± 0.40 ^c^	17.74 ± 0.20 ^b^	8.26 ± 0.10 ^c^	5.49 ± 0.20 ^c^
100 °C,15 min	69.18 ± 0.10 ^de^	16.06 ± 0.30 ^a^	9.60 ± 0.30 ^d^	5.16 ± 0.10 ^b^
Untreated	54.26 ± 0.10 ^A^	38.94 ± 0.10 ^F^	1.31 ± 0.10 ^A^	5.49 ± 0.20 ^E^
85 °C,10 min	65.62 ± 0.30 ^B^	23.13 ± 0.10 ^E^	7.44 ± 0.40 ^D^	3.81 ± 0.10 ^B^
85 °C,15 min	68.02 ± 0.10 ^C^	21.38 ± 0.20 ^D^	7.44 ± 0.20 ^D^	3.16 ± 0.10 ^A^
85 °C,20 min	75.89 ± 0.10 ^D^	16.47 ± 0.20 ^C^	4.29 ± 0.10 ^B^	3.35 ± 0.30 ^A^
85 °C,30 min	76.47 ± 0.20 ^E^	14.37 ± 0.30 ^A^	4.08 ± 0.20 ^BC^	5.08 ± 0.10 ^D^
85 °C,60 min	76.21 ± 0.10 ^DE^	14.82 ± 0.10 ^B^	4.64 ± 0.30 ^C^	4.33 ± 0.10 ^C^

Note: The same column of data letters in the table indicates the significant difference in the molecular weight distribution of protein molecules in the digested fluid of different heated temperature and time at the *p* < 0.05 level. Different letters indicate significant difference, and the same letter indicates that the difference is not significant.

**Table 3 foods-08-00594-t003:** The correlation between protein molecular distribution as a function of digestion SPIs and the DH at different heat treatments.

Molecular Mass Distribution/Da	Different Treatment Temperature	Different Time Processing
>100,000 Da	0.88 *	0.82 *
10,000–100,000 Da	−0.83 *	−0.79
3000–10,000 Da	0.80	0.36
<3000 Da	−0.54	−0.65

* Significantly different as compared to the correlation between protein molecular distribution and the DH at different heat treatments at 99% confidence level.

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
