# Peer review of "A Study of Structural Change during In Vitro Digestion of Heated Soy Protein Isolates"

_foods, 2019, doi:10.3390/foods8120594_

Round 1

Reviewer 1 Report

The manuscript entitled: “Study of stuctural change during in vitro digestion of heated Soy Protein Isolates” has been modified and improved. There are some minor remarks and criticisms to consider as detailed in the following. The authors should improve the Introduction by adding in the Introduction section some lines about the functional foods and nutraceuticals (see lines 32 and following) use of soybeans and some references in this regards should be added such as:

Santini, A.; Novellino, E. Nutraceuticals: shedding light on the grey area between pharmaceuticals and food. Expert Review of Clinical Pharmacology, 2018, 11(6), 545-547. https://doi.org/10.1080/17512433.2018.1464911.

Daliu, P.; Santini, A.; Novellino E. A decade of nutraceutical patents: where are we now in 2018? Expert Opin Ther Pat., 2018, 28(12): 875-882. doi: 10.1080/13543776.2018.1552260.

Patricia Daliu, Antonello Santini & Ettore Novellino. (2019). From pharmaceuticals to nutraceuticals: bridging disease prevention and management, Expert Review of Clinical Pharmacology, 12(1), 1-7. DOI: 10.1080/17512433.2019.1552135

The subparagraph 2.1. Materials and Chemicals should be implemented: please specify the reagents used and sources.

In paragraph 2.3 Heat treatment of SPI it is not clear if Authors followed the procedure as per the cited References or if there are modifications. If this is the case, please specify.

In the paragraph 2.2. Preparation of Soy Protein Isolates a scheme of the used procedure should be inserted to make it clearer for readers.

The advantages and the reason of using Raman Spectroscopy should be remarked in paragraph 2.6.

In paragraph 2.4. In Vitro Pepsin Digestion of HSPIs is not clear if Authors followed the procedure as per the cited References or if there are modifications. If this is the case, please specify.

The Results reported in Figure 2 and Table 1 in paragraph 3.2. “Protein Structure Determined by Raman Spectra” should be described and discussed in the text.

Paragraph 3.4.1. “Effects of Heating Temperature on Digestion Characteristics of SPI” should be better discussed. Figure 4 and 5 could be eliminated and observed results discussed in the text for better clarity.

Please follow the Journal guidelines for References and in the text with reference to capital names (se line 41) etc. and follow them.

In the Conclusion practical applications should be added as well the perspective point of view of the Authors. Please avoid repetition of what has been reported in the discussion of the results (see lines 304-305 and following).

Author Response

Dear Editor & Reviewers for Foods,

We would like to thank you for the opportunity to revise this manuscript for submission to Foods, and would further like to extend our thanks to the editor and reviewers for the valuable critiques and suggestions they have offered. We have now submitted the revised manuscript titled “A Study of stuctural change during in vitro digestion of heated Soy Protein Isolates” with modifications that have taken full consideration of the suggestions of the reviewers. A detailed point-by-point outline of how we have addressed each concern is presented below in this letter.  

This revised work is not under consideration for publication elsewhere and the authors have read and agreed to the contents of the manuscript that is currently being resubmitted. We hope this revised manuscript will satisfy the reviewers’ suggestions enough that it will now be found suitable for publication in Foods.

Sincere Regards,

Yang Li       

College of Food Science, Northeast Agricultural University, Harbin, 150030, China

Tel: 86-13613600153

The manuscript entitled: “Study of stuctural change during in vitro digestion of heated Soy Protein Isolates” has been modified and improved. There are some minor remarks and criticisms to consider as detailed in the following. The authors should improve the Introduction by adding in the Introduction section some lines about the functional foods and nutraceuticals (see lines 32 and following) use of soybeans and some references in this regards should be added such as:

Santini, A.; Novellino, E. Nutraceuticals: shedding light on the grey area between pharmaceuticals and food. Expert Review of Clinical Pharmacology, 2018, 11(6), 545-547. https://doi.org/10.1080/17512433.2018.1464911.

Daliu, P.; Santini, A.; Novellino E. A decade of nutraceutical patents: where are we now in 2018? Expert Opin Ther Pat., 2018, 28(12): 875-882. doi: 10.1080/13543776.2018.1552260.

Patricia Daliu, Antonello Santini & Ettore Novellino. (2019). From pharmaceuticals to nutraceuticals: bridging disease prevention and management, Expert Review of Clinical Pharmacology, 12(1), 1-7. DOI: 10.1080/17512433.2019.1552135

Response: Thank you for this kind mention. We have read your recommended references in detail and added some functional foods and health products related to soybeans in the introduction.

Line 32, “Additionally, SPI can reduce serum cholesterol and prevent heart and cerebrovascular diseases in the form of foods on the market (e.g. functional foods, pre and probiotics, food supplements, botanicals) and in pharmaceuticals. The bioavailability of nutrients affects its efficacy as a disease prevention agent. SPI has been suggested as an oil-water emulsion nanoparticle delivery system.”

The subparagraph 2.1. Materials and Chemicals should be implemented: please specify the reagents used and sources.

Response: We have specified the reagents used and sources in the subparagraph 2.1

Line 60, “Soybeans for this study were purchased from Hei Longjiang Agriculture Co., Ltd (Harbin, Hei Longjiang, China). Pepsin with an activity was 3,000 U/mg was purchased from Beijing Wohai Technology Ltd. (Beijing, China). An SDS-PAGE gel preparation kit was purchased from Beijing Suo Laibao Technology Co., Ltd (Beijing, China). All other chemicals were of analytical grade and were procured from Tianjin Chemical Reagent Co. (Tianjin, China).”

In paragraph 2.3 Heat treatment of SPI it is not clear if Authors followed the procedure as per the cited References or if there are modifications. If this is the case, please specify.

Response: In the SPI heat treatment experiment, we referred to the experimental method of Wang et al. The centrifugal speed was modified, which have been supplemented in paragraph 2.3.

Line 78, “The protocol for heat treatment of SPI was based on our previous research with slight modifications. Five g of SPI were first dissolved in 100 mL of phosphate buffered saline (PBS, 0.1 M, pH 7.4). The aqueous dispersion was stirred in a sealed glass tube and heated from 70 to 100 °C for 15 min, or heated at 85 °C for 10 to 60 min. The heated sample (HSPIs) was centrifuged at 9,500 rpm force for 20 min at 4 °C. After removing the insoluble compounds, the sample was cooled immediately in an ice bath for further experiments.”

In the paragraph 2.2. Preparation of Soy Protein Isolates a scheme of the used procedure should be inserted to make it clearer for readers.

Response: We have used a flow chart to make the preparation of soy protein Isolates is clearer for readers.

Line 74,

Figure 1. Flow chart of soybean protein isolate preparation.

The advantages and the reason of using Raman Spectroscopy should be remarked in paragraph 2.6.

Response: We have added the reason of using Raman Spectroscopy in paragraph 2.6.

Line 105, “Raman spectroscopy is a direct and a non-invasive technique which has been used to study the structure of SPI. Each spectrum of the sample was collected at 785 nm laser excitation wavelength, 300 mW laser power, 4 scans and 60 s exposure time. Baseline calibration was made for the average spectral data of samples scanned from Raman spectrophotometer (Renishaw, Gloucestershire, UK), and the phenylalanine band at 1,003 cm-1 was standardized. The Raman spectra of each sample were analyzed in the region of 400-3,100 cm-1. Each sample was scanned three times. After calculating the average value, the Raman spectra were drawn. The relative standard deviation was less than 5%.”

In paragraph 2.4. In Vitro Pepsin Digestion of HSPIs is not clear if Authors followed the procedure as per the cited References or if there are modifications. If this is the case, please specify.

Response: In the SPI heat treatment experiment, we referred to the experimental method of Chen et al. And made some modifications, which have been supplemented in paragraph 2.4.

An in vitro model that simulated sequential 184 gastric and intestinal digestion was applied to evaluate the influence of digestion on bioaccessibility 185 of curcumin (free, or encapsulated in the nanocomplexes with SPI) was accroding by Chen et al. We only conducted heat digestion SPI in vitro digestion experiment, and according to the experimental plan, 75 μL of 200 mM Na2CO3 (pH 11.0) was added to each mixture to stop the reaction by neutralization.

Line 85, “Pepsin digest of HSPI was conduct according to Chen et al. with slight modifications. Simulated gastric fluid (SGF) consisted of 3.2 mmol purified pepsin (3,000 units/mg protein, pH 1.2) containing 35 mmol NaCl. Pepsin solution was added drop-wise into SGF while vortexing for 5 min. The resulting solution was placed on ice. The concentrations of all test samples were 5% w/w of the HSPIs for SGF digestion assay. The in vitro gastric model consisted of a conical flask (100 mL) containing 10 mL of SGF-pepsin maintained at 37 °C with continuous shaking at 95 rpm. Aliquots (10 mL) were withdrawn into beakers 1 h at intervals during incubation, and 75 μL of 200 mmoL Na2CO3 (pH 11.0) was added to each mixture to stop the reaction by neutralization. The digestion was replicated in triplicate. After freeze-drying, the samples were stored at 4 °C until use.”

The Results reported in Figure 2 and Table 1 in paragraph 3.2. “Protein Structure Determined by Raman Spectra” should be described and discussed in the text.

Response: We further described and discussed Figure 2 of paragraph 3.2. And we modified the determination of protein structure by Raman spectroscopy in detail. We deleted Table 1 because only the secondary structure of protein was discussed in this experiment, so the data in Table 1 was not needed.

Line 170, “Analysis of the Raman spectra of proteins allow the unambiguous identification by spectral data base search algorithms. The Raman spectra of HSPIs are shown in Figure 3.”

Line 176, “Amide I and III are the most important bands in determining the secondary structure of proteins in several distinct vibration modes of the –CO–NH– amide. The main Raman characteristic peaks of the amide I band of the native SPI and HSPI were between 1,654 and 1,662 cm−1, indicating that the advantages of α-helices and random coil structures were presented (Figure 3). The amide III band at around 1,246 cm−1 (random coil) and 1,275 cm−1 (α-helix) was attributed to the in plane bending of N–H and stretching of C–N. The absorption at 1,305 cm−1 in the amide III region could be derived from a β-turn.”

Line 183, “Figure 3 shows a slight shift to lower frequencies of the intensity maximum of this band (1,660 cm−1) due to the heating process, indicating an increment in the α-helical structure due to thermal treatment.”

Paragraph 3.4.1. “Effects of Heating Temperature on Digestion Characteristics of SPI” should be better discussed. Figure 4 and 5 could be eliminated and observed results discussed in the text for better clarity.

Response: We have added the influence of heating temperature on the digestion characteristics of SPI has been further discussed in 3.4.1. And we have deleted Figure 4 and figure 5.

Line 307, “As shown in Table 3, although there was no correlation between SMW, LMW, and DH, the HMW content was significantly positively correlated with DH (P < 0.01, r = 0.88), and MMW content was significantly negatively correlated with DH (P < 0.01, r = -0.83). The results showed that with increasing temperature, the digestibility of soybean protein decreased, the content of high molecular weight peptide aggregates increased, and the content of medium and small molecular weight peptide increased. There was no correlation between MMW, SMW, LMW, and DH at different heating times, however HMW content showed a significant positive correlation with DH (P < 0.05, r = 0.82), indicating that protein digestibility decreased and the peptide content of the macromolecules in the digested products increased with extended durations of heating. This indicates that the formation of HMW protein aggregates is an important factor affecting the digestion of SPI.”

Please follow the Journal guidelines for References and in the text with reference to capital names (se line 41) etc. and follow them.

Response: The references and citations have be modified according to the Journal guidelines for References and in the text with reference to capital names.

In the Conclusion practical applications should be added as well the perspective point of view of the Authors. Please avoid repetition of what has been reported in the discussion of the results (see lines 304-305 and following).

Response: Thank you for this comment. According to the suggestions, we have re-written the conclusion part.

Line 318, “Food containing SPI is often affected by heat treatment throughout the course of production and processing, and heat treatment has a significant impact on the solubility, emulsification potential, and stability of SPI. In this paper, heat-treated SPI was studied by in vitro digestion assays in order to solve the structural changes of SPI in the human digestive system. Here, we have demonstrated that digestion of SPIs is greatly affected after exposure to heat. After the heating process, the subunit component of β-conglycinin of SPI was easier to digest and degrade compared with native SPIs. The digested products had a higher α-helix structure and a lower β-sheet structure. In summary, functional protein peptides with digestible, functional and antioxidant properties can be obtained by heat treatment. Moreover, heating enhanced both the emulsification and digestive utilization rate of SPI, indicating that the bioavailability of SPI can be improved. It can also effectively replace the nutritional and health effects of animal protein on human body. This study may provide valuable information on potential new avenues of application for SPI as a novel protein resource in the food industry.”

Reviewer 2 Report

In my opinion, the problem is gel filtration experiment, and then interpretation of the amount of MW fractions. First, the method is not described correctly (what was the elution solvent? what was the loaded sample volume? how was the amounts of protein calculated from chromatograms?). Second, detection at 280 nm is aromatic aminoacid specific, and depends on the amount of mostly Trp in the peptide. So, the same absorption of different peptides is not necessarily representation of the same amount.

Using the absorption at 230 nm is much better. 

In 2.8 paragraph, "approtionining" is, I believe, aprotinin. 

Also, other methodical details are missing, e.g. percentage of SDS PAGE gel.

Author Response

Dear Editor & Reviewers for Foods,

We would like to thank you for the opportunity to revise this manuscript for submission to Foods, and would further like to extend our thanks to the editor and reviewers for the valuable critiques and suggestions they have offered. We have now submitted the revised manuscript titled “A Study of structural change during in vitro digestion of heated Soy Protein Isolates” with modifications that have taken full consideration of the suggestions of the reviewers. A detailed point-by-point outline of how we have addressed each concern is presented below in this letter.  

This revised work is not under consideration for publication elsewhere and the authors have read and agreed to the contents of the manuscript that is currently being resubmitted. We hope this revised manuscript will satisfy the reviewers’ suggestions enough that it will now be found suitable for publication in Foods.

Sincere Regards,

Yang Li       

College of Food Science, Northeast Agricultural University, Harbin, 150030, China

Tel: 86-13613600153

In my opinion, the problem is gel filtration experiment, and then interpretation of the amount of MW fractions. First, the method is not described correctly (what was the elution solvent? what was the loaded sample volume? how was the amounts of protein calculated from chromatograms?).

Response: We have added the details of molecular weight distribution and marked them in the revised version.

Line 126, “Molecular weight distribution was determined according to the method of Yang et al. The samples were determined on an Agilent High Performance Liquid System and HiLoad 16/60 Superdex 200 prep grade pre-packed preparative columns with an Agilent UV detector set to 280 nm. Sample was eluted with mobile phase (0.02 M phosphate buffer containing 0.25 M NaCl (pH 7.2)) at a flow rate of 1 mL/min. A calibration curve of molecular weight was plotted from the average elution volume of the following standards: Cytochrome C (12,384 Da), aprotinin (6,500 Da), vitamin B12 (1,855 Da), oxidized glutathione (612 Da), and Glycine (75 Da) (Sigma Co., USA) were taken to make reference curve (r = 0.9955). A regression equation was established for the relative molecular mass (Mw) and elution volume (x) of Superdex 200 and Sephadex G-75. Aliquots of 20 μl samples were injected into the column. The molecular mass was estimated based on the elution time against those of molecular weight markers. The relative content of each peptide fraction was expressed as the percentage area of its chromatogram peak.”

Second, detection at 280 nm is aromatic aminoacid specific, and depends on the amount of mostly Trp in the peptide. So, the same absorption of different peptides is not necessarily representation of the same amount.Using the absorption at 230 nm is much better.

Response: Thank you for your suggestion and I'll take your suggestion seriously. But our experiment is based on our previous research basis to continue the in-depth study, so we chose the method of measuring the molecular weight of SPI at 280 nm. And we have consulted some references which is detected at 280 nm. Some of the documents we have found are attached below.

Fang Y , Zhang B , Wei Y , et al. Effects of specific mechanical energy on soy protein aggregation during extrusion process studied by size exclusion chromatography coupled with multi-angle laser light scattering[J]. Journal of Food Engineering, 2013, 115(2):220-225. Zhao J , Dong F , Li Y , et al. Effect of freeze–thaw cycles on the emulsion activity and structural characteristics of soy protein isolate[J]. Process Biochemistry, 2015:S1359511315300301. Dominguez-Vega E, Kotkowska O, Concepcion Garcia M, et al. Fast determination of the functional peptide soymetide in different soybean derived foods by capillary-high performance liquid chromatography [J]. JOURNAL OF CHROMATOGRAPHY A, 2011. In 2.8 paragraph, "approtionining" is, I believe, aprotinin.

Response: Thank you for your reminder. We have changed “approtionining” to “aprotinin”. (Line 131)

Also, other methodical details are missing, e.g. percentage of SDS PAGE gel.

Response: We have added the methodical details of using SDS-PAGE in paragraph 2.7, the methodical details of using In Vitro Pepsin Digestion of HSPIs in paragraph 2.4 and Raman Spectroscopy in paragraph 2.6.

Line 114, “Sodium dodecyl sulphide-polyacrylamide gel electrophoresis (SDS-PAGE) was performed according to the method of Laemmli et al. Preparation for the 5% stacking gel and 12% separating gel was according to the specification provided by Suo Laibao Technology (Beijing, China). Tweenty μL protein samples or their digestive products were mixed in a buffer (1:1, v/v) containing β -mercaptoethanol in an equal volume and then denatured inboiling water for 5 min. The samples were cooled at room temperature, and then 10 μL aliquots of the samples (5 mg/mL) were loaded into the gel. Electrophoresis was carried out at 80 mV in the stacking gel and then separated in the gel under 120 mV. After about 3 hours of electrophoresis, coomassie brilliant blue was stained with R-250 for 12 hours until the dye reached the bottom of the gel. After electrophoresis, the gel was stained with 0.05% coomassie brilliant blue R-250 solution until a clear background appeared. We used a Molecular Imager Gel Doc (Bio-Rad Laboratories, California, USA) to capture the gel. ”

Line 85, “Pepsin digest of HSPI was conduct according to Chen et al. [21] with slight modifications. Simulated gastric fluid (SGF) consisted of 3.2 mmol purified pepsin (3,000 units/mg protein, pH 1.2) containing 35 mmol NaCl. Pepsin solution was added drop-wise into SGF while vortexing for 5 min. The resulting solution was placed on ice. The concentrations of all test samples were 5% w/w of the HSPIs for SGF digestion assay. The in vitro gastric model consisted of a conical flask (100 mL) containing 10 mL of SGF-pepsin maintained at 37 °C with continuous shaking at 95 rpm. Aliquots (10 mL) were withdrawn into beakers 1 h at intervals during incubation, and 75 μL of 200 mmoL Na2CO3 (pH 11.0) was added to each mixture to stop the reaction by neutralization. The digestion was replicated in triplicate. After freeze-drying, the samples were stored at 4°C until use.”

Line 105, “Raman spectroscopy is a direct and a non-invasive technique which has been used to study the structure of SPI. Each spectrum of the sample was collected at 785 nm laser excitation wavelength, 300 mW laser power, 4 scans and 60 s exposure time. Baseline calibration was made for the average spectral data of samples scanned from Raman spectrophotometer (Renishaw, Gloucestershire, UK), and the phenylalanine band at 1,003 cm-1 was standardized. The Raman spectra of each sample were analyzed in the region of 400-3,100 cm-1. Each sample was scanned three times. After calculating the average value, the Raman spectra were drawn. The relative standard deviation was less than 5%.”

This manuscript is a resubmission of an earlier submission. The following is a list of the peer review reports and author responses from that submission.

Round 1

Reviewer 1 Report

Authors presented interesting experimental results of soy protein isolate digestion in function of heat treatment and time of digestion. The important input of the manuscript is that authors provide also explanations and mechanisms which may led to the obtained results. However, before the manuscript can be considered for publication the following points should clarified.

Major points:

Line 74. Please provide exactly which buffer was used. Line 150. Please clarify how long the temperature was applied in case of data presented in Fig. 1A and at what temperature data on Fig. 1B were obtained. Maybe authors should think about surface plot or other way of presenting the data, if they have more results in the experimental domain of DH=f(temperature, time). Table 2 and 3. Clarify what all uppercases exactly mean. That clarification might influence the discussion section. Line 310-312. Why authors mention in Discussion "emulsifying utilization" if they did not investigate that? Discussion section. Authors should clearly state what is the novelty of their research, in comparison to other, for example to reference 43.  

Minor points:

Careful English check. For example, line 31-32, sentence “As well….”. Table 1. Reorganize table in the way that words and abbreviations would not be divided between lines. Line 167. Insert space between Figure and 1. Line 189. AB-globulin ? Line 229. Strange commas between numbers representing temperatures. Line 243. Change KDa to kDa. Improve readability of the figures 4 and 5. Table 4. For what does star stand for?

Author Response

请参阅附件

Reviewer 2 Report

This manuscript describes effects of heat treatments on pepsin digestion, protein structure, and aggregates of soy protein isolate. Overall, the authors did not follow the basic rules of writing a scientific paper. For examples, they discussed in the Results section and did without any references in the Discussion section. Legends of Figures are not adequate that the readers would not be able to understand. This paper also needs English editing.

Other scientific problems were mentioned as following:

1. The authors showed the degree of hydrolysis (DH) of SPI treated at different heating temperature and time. The difference of DH was up to ONLY 4% and experimental conditions were limited.

2. The authors investigated the effect of heating temperature and time on the secondary structural changes of SPI by Raman spectroscopy. There were a little structural differences in SPI treated at different heating temperature and time. This is reasonable because denatured structure at a high temperature reveals structural reversibility at a room temperature, indicating that they could not show the structural differences at different conditions.

Reviewer 3 Report

The manuscript entitled: "The Mechanism of Heat Treatment on improving In Vitro Digestion of SPI" reports data on soy protein isolates. The manuscript is in general poorly assessed, nonetheless there a few criticisms;some arereported in the  following.

First of all please avoid the use of acronyms in the title of the manuscript (e.g. see SPI in the title:please use full definition and reduce the title lenght). As an example you might use the following title: "Heat treatment improves the in vitro digestion of soy protein isolates" or something like that. Experimental part is confused and it should be rewritten to clearly assess the methods and results. It is not clear the "mechanism" as per the manuscript title. The scope and possible outcome of the experimental work should be assessed and cleared better in the manuscript as well as the nutraceutical potential of soy protein isolates. Some criticisms are in the following:

Line 67 and 72 paragraph 2.2:please correct centrifuge speed;

Line 74: what does it mean "standard buffer"? please clarify;

Paragraph 2.4. What does that mean "appropriate amount": please give details of the experimental procedures adopted;

Paragraph 2.6: authors report standard deviation and a reference: please explain. The reported 5% is a literature data? This should be cleared;

Figures are not clear and should be redrawn for proper readability. As they are not clear, it is difficult to interpret the results.  How the selected temperature range has been selected? Please describe the criteria used. For this reason it is suggested to better assess the experimental part and the discussion before resubmitting the manuscript.

Conclusion section is missing while the last paragraph entitled "Discussion" is short and similar to Conclusion section. This should be better assessed as well as the perspective point of view of the Authors, and any possible practical application of the obtained results. This is only mentioned.

Some of the literature references are too old: please consider he recent literature available in the area of interest to cite. English full revision of the English text is necessary: please do it.

Round 2

Reviewer 1 Report

Authors answered to all questions. The manuscript can be published.

Reviewer 2 Report

I believe that this revised manuscript warrants publication in Foods. I pointed out in my previous review that the degree of hydrolysis (DH) of heat-treated soy protein isolate (SPI) was up to ONLY 4% compared with unheated SPI in their experimental system. This means that more than 90% of heated SPI was not affected by enzyme treatments. However, the authors responded that heat treatment had a significant effect on digesting SPI. Thus, their answer does not make sense. Another my comment was that their results analyzed by Raman spectroscopy showed a little structural difference in heat-treated SPI. Their answer was that digestion had little effect on the secondary structure of untreated and heat-treated proteins but had great influence on the molecular weight of proteins. However, the authors used a chromatography to determine the molecular weights of heat and enzyme-treated SPI. MORE THAN 50% of heat and enzyme-treated SPI eluted in VOID volume, indicating that they failed to characterize the molecular weights. We cannot get information on the molecular weights of proteins from void volume.

Reviewer 3 Report

The manuscript entitled "The Mechanism of Heat Treatment on improving In Vitro Digestion of SPI" ha sbeen modified, nonetheless is still unclear and poorly written. Figure 4 is not mentioned in the text. Moreover they seem to ne not necessary together with Figure 5; they could be deleted and the observations reported adding a sentence in the text. Authors should check carefully the consistency of Tables and Figures with which is reported in the text paragraphs. It requires revision of English for readability. A Discussion section should be included avoiding too many sub-paragraphs.